# Analysis of the Expression Profile in COVID-19 Patients in the Russian Population Considering Disease Severity, Mortality, and Cytokine Storm

**DOI:** 10.3390/biomedicines13040863

**Published:** 2025-04-03

**Authors:** Valentin Shimansky, Oleg Popov, Alexander Kel, Igor Goryanin, Tatiana Klochkova, Svetlana Apalko, Natalya Sushentseva, Anna Anisenkova, Sergey Mosenko, Sergey Shcherbak

**Affiliations:** 1The Saint Petersburg State Health Care Establishment the City Hospital No. 40, 197348 Sestroretsk, Russia; ospopov@outlook.com (O.P.); tklochkova@list.ru (T.K.); svetlana.apalko@gmail.com (S.A.); navicula@yandex.ru (N.S.); anna_anisenkova@list.ru (A.A.); neurologist@mail.ru (S.M.); b40@zdrav.spb.ru (S.S.); 2Medical Institute, St. Petersburg State University, 199034 Saint Petersburg, Russia; 3Department of Research & Development, GeneXplain GmbH, 38302 Wolfenbüttel, Germany; alexander.kel@genexplain.com; 4School of Informatics, University of Edinburgh, Edinburgh EH8 9YL, UK; igor.goryanin@ed.ac.uk

**Keywords:** gene expression, RNA sequencing, COVID-19

## Abstract

**Background/Objectives**: The COVID-19 pandemic has posed a significant challenge to global healthcare systems and has prompted a need for a better understanding of the molecular mechanisms underlying SARS-CoV-2 infection. This study aims to analyze differential gene expression in COVID-19 patients to identify regulatory genes influencing key pathways involved in disease progression. **Methods**: We conducted a transcriptomic analysis of patients admitted to the Infectious Disease Department of City Hospital No. 40, confirmed with SARS-CoV-2 via PCR. The study received ethical approval (protocol No. 171, 18 May 2020), and all participants provided informed consent. Total RNA was extracted from blood samples, followed by RNA sequencing using the DNBSEQ-G400 platform. Differential gene expression was analyzed using the Mann–Whitney test, and Gene Ontology enrichment analysis was performed to identify relevant biological processes. **Results**: Our analysis revealed significant number of differentially expressed genes within studied groups (severity, outcome, cytokine storm and paired samples). These genes are involved in key regulatory and signal transduction pathways governing immune responses, intercellular communication, and the metabolism of various compounds. Furthermore, we identified genes *ALOX15*, *PRL*, *FLT3*, *S100A8*, *S100A12*, *IL4*, *IL13*, and a few others as master regulators within the studied pathways, which represent promising candidates for further investigation as potential therapeutic targets. **Conclusions**: This study highlights critical gene expression changes associated with COVID-19 severity and outcomes, identifying potential biomarkers. Our findings contribute to the understanding of the molecular drivers of COVID-19 and suggest new avenues for therapeutic interventions aimed at modulating immune responses.

## 1. Introduction

The COVID-19 pandemic has posed a significant challenge to healthcare systems worldwide. Extensive research has been conducted to understand the disease from various perspectives, with a particular focus on the impact of SARS-CoV-2 infection on the human transcriptome. This area of research is critical for elucidating the molecular mechanisms that drive disease progression and for identifying potential therapeutic targets. Among the methods employed, RNA sequencing (RNA-seq) has become a cornerstone due to its ability to provide comprehensive data on gene expression across numerous patients in a relatively short time frame.

RNA-seq studies have revealed that SARS-CoV-2 infection causes profound alterations in the transcriptome of blood cells, with changes in gene expression that correlate with disease severity. For example, a study identified 2,289 genes that were upregulated and 912 genes that were downregulated in the blood of COVID-19 patients across all severity levels compared to controls [1]. These changes predominantly affect genes involved in immune function, with a gradual decrease in the expression of genes associated with adaptive immunity as the severity of the disease increases. Concurrently, genes involved in innate immunity exhibit increased expression [2].

Further research has identified specific molecular mechanisms disrupted by SARS-CoV-2. For instance, a study focused on protein ubiquitination in COVID-19 identified 268 differentially expressed genes in peripheral blood mononuclear cells of patients with severe disease, highlighting key transcription factors and microRNAs involved in ubiquitination regulation [3]. Similarly, the differential expression of mRNAs, microRNAs, and long non-coding RNAs has been observed, with overexpressed mRNAs in COVID-19 patients being primarily involved in antigen processing and T-cell-mediated cytotoxicity, while downregulated mRNAs are linked to glycogen biosynthesis [4].

Age-related changes in the blood transcriptome have also been documented, revealing that certain genes involved in immune response, inflammation, and cell adhesion exhibit altered expression profiles with aging, with distinct differences observed between men and women [5]. Interestingly, the transcriptomic signature of COVID-19 shows significant overlap with that of other acute respiratory infections, such as influenza [6].

Recent studies have begun to explore the dynamic changes in the blood transcriptome during the course of COVID-19 and recovery. Notably, it has been observed that gene expression levels do not revert to those of healthy individuals even months after clinical recovery [7]. As a result, efforts are underway to develop transcriptome panels for profiling the immune response to SARS-CoV-2, with target genes categorized by their immunologic relevance, role in disease progression, and interaction with viral structures [8].

Given the insights provided by transcriptomic analysis, this study aims to analyze differential gene expression in COVID-19 patients to identify regulatory genes that influence key molecular and cellular pathways involved in the pathogenesis of the disease. The study was conducted with the approval of the expert ethics board of City Hospital No. 40 (protocol No. 171, dated 18 May 2020), and all patients provided informed consent. Biomaterial from the biobank collection of City Hospital No. 40 was utilized to achieve the study objectives.

## 2. Materials and Methods

### 2.1. Participant Characteristics

This study involved patients from the Infectious Disease Department of City Hospital No. 40, who were admitted for treatment of SARS-CoV-2 infection, confirmed via polymerase chain reaction (PCR). All participants provided informed voluntary consent in accordance with ethical standards. The study protocol was approved by the Expert Ethics Board of City Hospital No. 40 (protocol No. 171, dated 18 May 2020).

Biological samples were obtained from the hospital’s biobank collection, and these samples were used to achieve the study’s objectives. The patient cohort was stratified into several groups based on the following criteria:Disease severity: Patients were categorized into severity 1 group (mild-to-moderate disease course) and severity 2 group (severe-to-extremely severe disease course).Disease outcome: This classification distinguished between patients with different clinical outcomes: lethal group and non-lethal group.Cytokine storm: Patients were assessed for the presence of a cytokine storm at the time of sample collection.Paired samples: A subset of patients had paired samples collected, representing both the healthy state and post-SARS-CoV-2 infection.

The demographic characteristics, including the age and sex (M for male patients and F for female patients) distribution of these groups, are summarized in Table 1, Table 2, Table 3 and Table 4. Additionally, Table 1 and Table 2 present comorbidity statistics in the form of the Charlson comorbidity index, which was calculated for the following list of comorbidities: history of myocardial infarction, congestive heart failure, peripheral arterial disease, cerebrovascular disease, dementia, chronic lung disease, connective tissue disease, peptic ulcer disease, diabetes mellitus, kidney damage, hemiplegia or paraplegia, leukemia, lymphoma, malignancy, liver damage, and AIDS.

### 2.2. RNA Extraction and Sequencing

Total RNA was manually extracted from blood samples using the phenol-chloroform extraction method, following established protocols. To prepare sequencing libraries, we utilized the KAPA RiboErase HMR kit (Kapa Biosystems, Wilmington, MA, USA) for ribosomal RNA (rRNA) depletion, followed by the KAPA RNA HyperPrep kit (Kapa Biosystems, Wilmington, MA, USA) for library preparation. For adapter ligation, we employed the KAPA Universal Adapter (Kapa Biosystems, Wilmington, MA, USA) in combination with the KAPA UDI Primer Mixes (Kapa Biosystems, Wilmington, MA, USA), ensuring compatibility and efficiency in subsequent sequencing steps. The conversion of libraries to a format suitable for high-throughput sequencing was achieved using the High-Throughput Sequencing Primer Kit (App-C) (MGI Tech Co., Ltd., Shenzhen, China) and the MGIEasy Universal Library Conversion Kit (App-A) (MGI Tech Co., Ltd., Shenzhen, China). Sequencing was conducted on an DNBSEQ-G400 platform, utilizing 100 base pair paired-end reads. The sequencing run was performed on a DNBSEQ-G400 High-throughput Sequencing Set (PE100, 360 GB) cell (MGI Tech Co., Ltd., Shenzhen, China), allowing for comprehensive coverage and high data yield.

### 2.3. Bioinformatics Analysis

#### 2.3.1. Quality Control and Alignment

The quality control of the raw sequencing reads was conducted using the FastQC tool [9], which provides an overview of the data quality and identifies any potential issues that may affect downstream analysis. Following quality assessment, reads were aligned to the human reference genome (hg38) using the STAR aligner [10]. STAR was configured with default parameters, optimizing both speed and alignment accuracy, making it well suited for processing large RNA-seq datasets.

#### 2.3.2. Read Counting and Gene Annotation

Post-alignment, read counting and gene annotation were performed using the featureCounts tool v2.0.3 [11]. This software enables the efficient and accurate quantification of gene expression levels, facilitating downstream statistical analyses.

#### 2.3.3. Statistical Analysis

Differential gene expression was assessed using the nonparametric Mann–Whitney test to determine statistically significant differences in gene expression between groups. A nonparametric test was chosen because neither of the read count distributions was normal, which was checked with the Shapiro–Wilk normality test. Log fold change (logFC) was employed as a measure of differential expression, providing a quantitative estimate of expression differences across conditions. All statistical analyses were carried out in the R statistical software environment.

#### 2.3.4. Gene Ontology and Pathway Enrichment Analysis

Gene Ontology (GO) pathway enrichment analysis was performed to identify biological processes, cellular components, and molecular functions that were overrepresented among differentially expressed genes. This analysis, along with master regulator analysis, was conducted using the Genome Enhancer tool on the geneXplain platform. The Genome Enhancer tool enables the identification of key regulatory elements and pathways by integrating gene expression data with known biological networks.

## 3. Results

Our analysis revealed significant alterations in the blood transcriptome of COVID-19 patients, with notable differences observed across groups categorized by disease severity, outcome, and the presence of a cytokine storm. Initially, we compared gene expression levels between patients with mild and severe disease courses. A total of 4734 genes were identified as differentially expressed, surpassing the statistical significance threshold of *p* < 0.01. Among these, 806 genes were upregulated and 3925 genes were downregulated in the severe disease group relative to the mild disease group.

Subsequently, we analyzed gene expression based on disease outcome, differentiating between patients with lethal and non-lethal outcomes. This analysis identified 9869 differentially expressed genes, with 348 genes showing upregulation and 9521 genes showing downregulation in patients with a lethal outcome compared to those with a non-lethal outcome (Figure 1).

Lastly, we examined gene expression changes in relation to the presence of a cytokine storm. In this comparison, 290 genes were found to exhibit significant alterations in expression. Specifically, 39 genes were upregulated and 251 genes were downregulated in the group of patients who experienced a cytokine storm compared to those who did not. The detailed results of these analyses are presented in the corresponding Appendix A tables. The top ten differentially expressed genes for each group are presented in Appendix A Table A1, Table A2 and Table A3.

To gain a more comprehensive understanding of the molecular mechanisms impacted by SARS-CoV-2 infection, we performed Gene Ontology (GO) enrichment analysis on differentially expressed genes identified in our study. This analysis was conducted across groups stratified by disease severity, outcome, and the presence of a cytokine storm.

### 3.1. GO Enrichment Analysis by Disease Severity

In groups categorized by disease severity, the GO enrichment analysis revealed that genes with increased expression were predominantly associated with terms related to adaptive immune response, signaling, intercellular communication, and other immune-related processes. Conversely, genes with decreased expression were primarily involved in processes related to cellular nitrogen metabolism, heterocyclic compound metabolism, and general gene expression regulation. The top GO terms identified in this study are presented in Appendix A Table A4 and Table A5.

### 3.2. GO Enrichment Analysis by Disease Outcome

For the groups categorized by disease outcome, our analysis identified that upregulated genes were enriched in pathways associated with neutrophil degranulation, neutrophil-mediated immunity, and responses to external stimuli. Downregulated genes were largely associated with metabolic pathways involving cellular nitrogen compounds, nucleobase-containing compound metabolism, and heterocyclic compounds. The top GO terms identified in this study are presented in Appendix A Table A6 and Table A7.

### 3.3. GO Enrichment Analysis in the Context of Cytokine Storm

In the context of diagnosed cytokine storm, the GO enrichment analysis showed that upregulated genes were enriched in pathways responsible for neutrophil degranulation, neutrophil-mediated immunity, and response to external stimuli. On the other hand, downregulated genes were enriched in pathways associated with adaptive immune response, the negative regulation of apoptosis in bone marrow cells, and the positive regulation of isotype switching in interferon-producing cells. The top GO terms identified in this study are presented in Appendix A Table A8 and Table A09.

### 3.4. Paired Sample Analysis

Additionally, we analyzed a unique set of paired samples from eight patients, collected both in a healthy state and following confirmed SARS-CoV-2 infection. This analysis identified 24 differentially expressed genes. The GO enrichment analysis of these genes indicated significant involvement in the type I interferon signaling pathway and cellular responses to type I interferon. The top three GO terms identified in this analysis are presented in the accompanying Figure 2, Figure 3 and Figure 4.

To elucidate the gene regulatory and signal transduction mechanisms underlying the extensive gene expression differences observed among patients with varying responses to SARS-CoV-2 infection, we conducted an upstream network analysis using the Genome Enhancer tool. This approach allowed us to identify key regulatory elements and pathways that may be driving these expression changes.

Specifically, we performed a master regulator analysis across patient groups stratified by disease severity, clinical outcome, and the presence of a cytokine storm. This analysis aimed to identify central regulatory genes or proteins that could explain the observed patterns of differential gene expression. However, due to the limited number of differentially expressed genes in the paired samples, this type of analysis could not be effectively performed for that subset. The results of the master regulator analysis for the various patient groups are detailed in Table 5. The summary scheme of the master regulators and the major metabolic processes in which they are involved is illustrated in Figure 5. Corresponding master regulator network figures for each group are presented in Appendix A Figure A1, Figure A2, Figure A3, Figure A4, Figure A5 and Figure A6.

## 4. Discussion

Understanding the interplay between gene regulatory mechanisms and signal transduction pathways in both healthy and disease states is crucial for identifying potential therapeutic targets in clinical settings [12]. In this study, we identified differentially expressed genes involved in pathways such as immune response regulation, nitrogenous base metabolism, signal transduction, and non-coding RNA processing. Through our analyses, we reconstructed gene regulatory and signaling networks, identifying key master regulators that play pivotal roles in these processes.

One such master regulator, *ALOX15*, exhibited reduced expression in groups with lethal outcomes and those experiencing cytokine storms. The ALOX15 gene encodes an enzyme known as lipoxygenase, which plays a crucial role in the metabolism of polyunsaturated fatty acids. Its primary functions include inflammation regulation, cellular signaling, lipid metabolism, and immune response [13,14]. This finding suggests a persistent inflammatory response, with a disrupted resolution mechanism, potentially contributing to more severe disease outcomes [15]. Given its role as a master regulator, *ALOX15* emerges as a promising candidate for further investigation as both a biomarker and a potential therapeutic target.

Additionally, the reduced expression of the *IL4* gene was observed in groups with fatal outcomes and cytokine storms. *IL4* is a multifunctional cytokine critical for the regulation of immune responses. It also mediates the demethylation of histone H3 trimethyl-lysine 27 (H3K27me3) at the promoter region of the *ALOX15* gene, thus inducing its transcription [16]. Although no direct association between *IL4* and severe disease progression has been conclusively established in the literature, its role as a regulatory gene warrants further investigation. The potential to modulate the inflammatory response through *IL4*-*ALOX15* interaction could lead to improved disease outcomes [17].

Furthermore, *KNDC1* was downregulated in groups with severe disease and fatal outcomes, as well as in paired sample analyses. This gene is implicated in signaling pathways, protein recognition, and functional regulation, and has been identified by other researchers as a significant marker when comparing recovered and severe COVID-19 patient samples [18]. In groups with fatal outcomes and cytokine storms, the downregulation of genes encoding integrin subunits was also noted, suggesting increased susceptibility to viral infection. Similarly, the *PRL* gene, involved in adaptive immunity and inflammatory processes, showed reduced expression in these groups.

Our study also identified significant enrichment of overexpressed genes involved in metabolic pathways related to neutrophil degranulation and neutrophil-mediated immune responses in groups diagnosed with cytokine storms and lethal outcomes. Prior research has shown that dysregulation in neutrophil function and infiltration into inflamed tissues can impair the overall immune response, exacerbating the severity of the disease [19]. These observations suggest that the direct impact of COVID-19 on neutrophils may be a contributing factor to these metabolic abnormalities.

In addition, our analysis revealed that pathways responsible for the regulation of the adaptive immune response were enriched with upregulated genes in groups with severe disease and cytokine storms. The differential expression analysis in paired samples highlighted the involvement of genes in antiviral defense processes, particularly the interferon-alpha (interferon-1) and interferon-gamma signaling pathways. These findings indicate multiple disruptions in the metabolic processes of the organism.

The *PRL* gene, identified as a master regulator in outcome-based analyses, possesses both pro- and anti-inflammatory functions and plays a role in the regulation of immune cell function. This makes it an attractive target for further detailed research [20]. The master regulator analysis also identified a cluster of cytokine-related genes, including *FLT3*, *S100A8*, *S100A12*, *IL4*, and *IL13*, which directly encode cytokines. These cytokines are involved in the regulation of antiviral responses [21]. The S100 proteins, produced in granulocytes, monocytes, and macrophages, are integral to both innate and adaptive immunity. Moreover, *KIT* and *FLT3*, which are cytokine receptors, belong to the type 3 protein-tyrosine kinase receptor family [22]. The *MMP2*, *MMP9*, and *MAF* genes, which regulate cytokines, encode matrix metalloproteinases that perform various regulatory functions in immune and inflammatory processes [23]. Additionally, the proteins encoded by *KLRC1* and *KLRD1* are part of the NK-cell receptor family, while *MAPK3* and *MAPK8* are involved in the regulation of the MAP-kinase cascade. The master regulators identified in our study are implicated in signal transduction metabolic pathways with dysregulated components that are associated with more severe disease courses and increased mortality, making them promising targets for further investigation [24].

Even though our study provides some promising results, some limitations of the methods used should be taken into consideration. Biological variability among samples can introduce noise, complicating the identification of genuine biological signals, and this variability may arise from individual physiological differences or environmental factors. Furthermore, while Gene Ontology and pathway enrichment analyses offer valuable insights, they may not fully capture the complexity of biological processes, and the overrepresentation of certain pathways should be interpreted with caution regarding their functional relevance. Also, due to the computational nature of our master regulator predictions, additional experimental validation should be considered.

## 5. Conclusions

In this study, we performed a comprehensive transcriptomic analysis of COVID-19 patients from the Russian population residing in the northwest region of the country. Key findings from our study highlight specific alterations in gene expression that are associated with the severity and lethality of COVID-19. We discovered several potential biomarker genes related to this, among which are the genes *ALOX15*, *IL4*, *KNDC1*, and integrin subunits. In the course of our research, we identified master regulators within the gene expression networks, which provides promising avenues for future research, with the goal of refining our understanding of the molecular drivers of COVID-19 and exploring new therapeutic interventions.

## Figures and Tables

**Figure 1 biomedicines-13-00863-f001:**
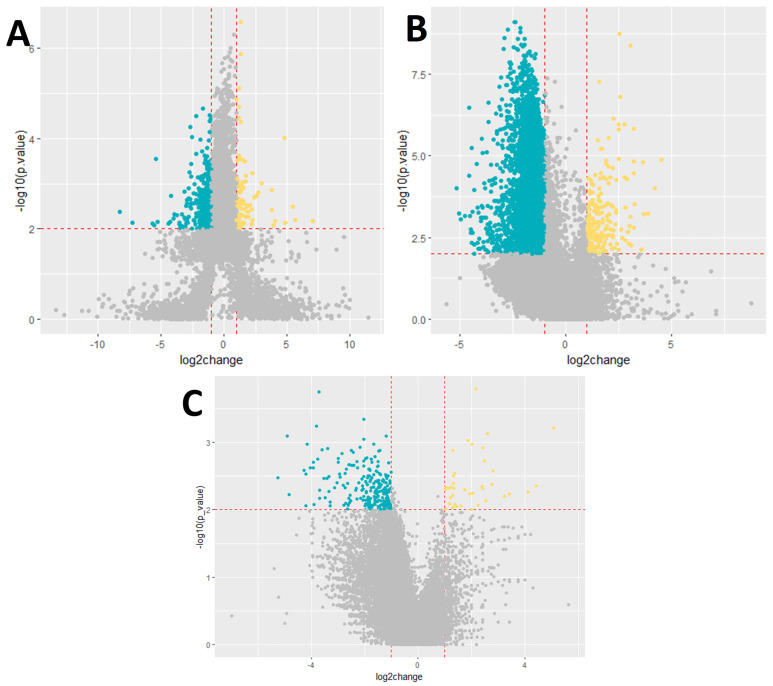
Volcano plot showing the differential expression of genes. (**A**) Severity, (**B**) outcome, and (**C**) cytokine storm. Dark blue−downregulated genes; *p*-value < 0.01; log2foldchange < −1. Yellow−upregulated genes; *p*-value < 0.01; log2foldchange > 1.

**Figure 2 biomedicines-13-00863-f002:**
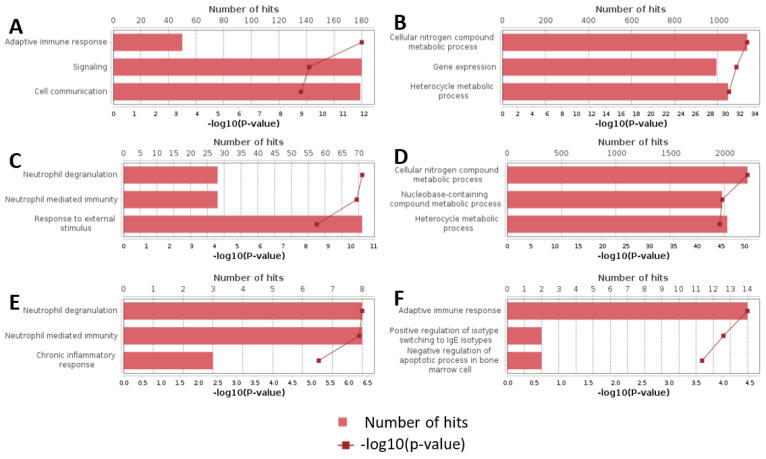
Top results of GO enrichment analysis. (**A**) Upregulated genes in severity groups. (**B**) Downregulated genes in severity groups. (**C**) Upregulated genes in outcome groups. (**D**) Downregulated genes in outcome groups. (**E**) Upregulated genes in cytokine storm groups. (**F**) Downregulated genes in cytokine storm groups.

**Figure 3 biomedicines-13-00863-f003:**
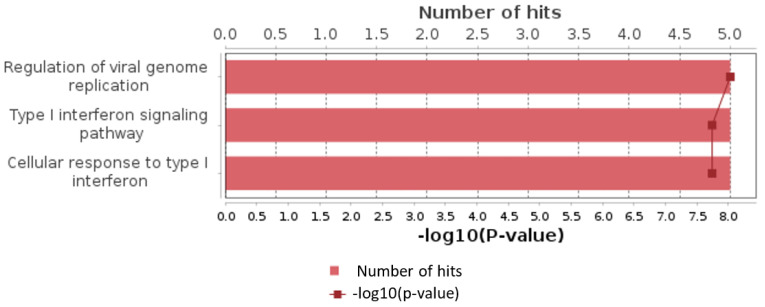
Top results of GO enrichment analysis for paired samples group.

**Figure 4 biomedicines-13-00863-f004:**
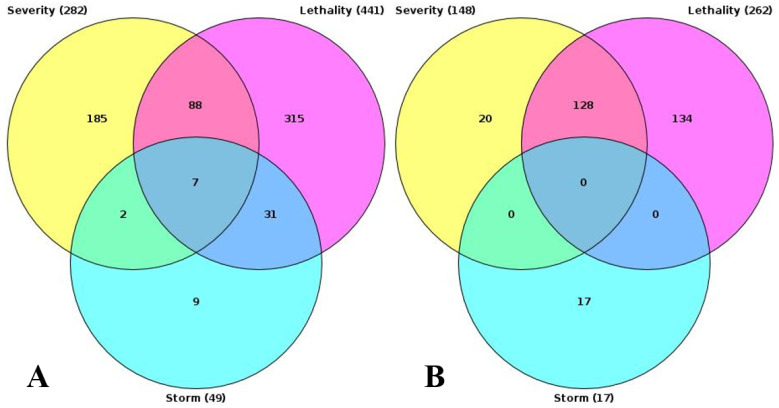
Intersection of results of GO enrichment analysis. (**A**) GO terms, enriched with upregulated genes. (**B**) GO terms, enriched with downregulated genes.

**Figure 5 biomedicines-13-00863-f005:**
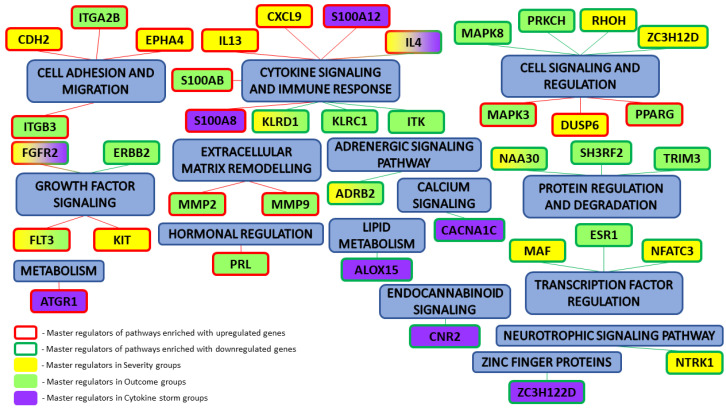
Summary results of master regulator analysis with the corresponding metabolic processes that they are involved in.

**Table 1 biomedicines-13-00863-t001:** Age–sex structure and comorbidity statistics of groups divided by disease severity.

Severity	Number of Patients	Number, M	Age, M	Number, F	Age, F	Charlson > 0	Mean Charlson
Severity 1	109	54	60.9 ± 13.3	55	60.7 ± 13.6	77	4.01
Severity 2	37	21	59.4 ± 13.8	16	60.9 ± 13.5	33	4.48

**Table 2 biomedicines-13-00863-t002:** Age–sex structure and comorbidity statistics of groups divided by disease outcome.

Outcome	Number of Patients	Number, M	Age, M	Number, F	Age, F	Charlson > 0	Mean Charlson
Non-Lethal	120	57	58.5 ± 12.5	63	58.1 ± 13.7	85	3.95
Lethal	26	18	69.7 ± 9.1	8	78.0 ± 8.7	25	4.84

**Table 3 biomedicines-13-00863-t003:** Age–sex structure of the groups divided by cytokine storm at the time of sample collection.

Cytokine Storm	Number of Patients	Number, M	Age, M	Number, F	Age, F
Storm	12	8	57.2 ± 11.7	4	58.1 ± 12.2
No storm	32	13	59.5 ± 12.1	19	59.4 ± 12.0

**Table 4 biomedicines-13-00863-t004:** Age–sex structure of paired-samples group.

Group	Number of Patients	Number, M	Age, M	Number, F	Age, F
Paired	8	7	73.0 ± 9.5	1	83 ± 0.0

**Table 5 biomedicines-13-00863-t005:** Summary table of master regulator analysis results.

Severity		Outcome		Cytokine Storm	
Increased Expression	Decreased Expression	Increased Expression	Decreased Expression	Increased Expression	Decreased Expression
*CDH2*	*ADRB2*	*FLT3*	*ADRB2*	*ATGR1*	*ALOX15*
*CXCL9*	*FXN*	*ITGA2B*	*ERBB2*	*S100A12*	*CACNA1C*
*DUSP6*	*KLRDI*	*ITGB3*	*ESR1*	*S100A8*	*CNR2*
*EPHA4*	*MAF*	*MAPK3*	*ITK*		*FGFR2*
*FGFR2*	*NAA30*	*MMP2*	*KLRCI*		*IL4*
*FLT3*	*NFATC3*	*MMP9*	*KLRDI*		
*IL13*	*NTRKI*	*PPARG*	*MAPK8*		
*IL4*	*RHOH*	*PRL*	*NAA30*		
*KIT*	*ZC3H12D*	*S100AB*	*PRKCH*		
			*PRNP*		
			*SH3RF2*		
			*TRIM3*		
			*ZC3H122D*		

## Data Availability

Personal genetic and clinical data are under restrictions and are available through collaboration with City Hospital No. 40.

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
