# Peer review of "Analysis of the Expression Profile in COVID-19 Patients in the Russian Population Considering Disease Severity, Mortality, and Cytokine Storm"

_biomedicines, 2025, doi:10.3390/biomedicines13040863_

Round 1
Reviewer 1 Report
Comments and Suggestions for Authors
The work by Valentin Shimansky et al., entitled "Expression profile analysis of COVID-19 patients" has important information in the context of Russia. This should be used to make a comparison with similar works from other countries.
I think that the following points should be improved.
- L1-11.-What genes, please improve the abstract
- Authors should indicate the meaning of all acronyms when they first appear.
- Please provide citation for all methods used.
- L58.- What morbidity did the patients studied have?
- Limitations are missing. The paper does not include a section discussing the limitations of the study.
- The authors must present a model that integrates the regulatory keys and signal transduction of the immune response, intercellular communication and metabolism.
Author Response
Response to Reviewer 1 Comments
|
||
1. Summary |
|
|
Thank you very much for taking the time to review this manuscript. Please find the detailed responses below and the corresponding revisions and corrections in the re-submitted files.
|
||
2. Point-by-point response to Comments and Suggestions for Authors |
||
Comments 1: L1-11.-What genes, please improve the abstract |
||
Response 1: Thank you for the comment, corresponding addition of a short list of master-regulator genes that we identified was made in L1-11. |
||
Comments 2: Authors should indicate the meaning of all acronyms when they first appear. |
||
Response 2: Thanks, I’ve done the corrections in text. |
||
Comments 3: Please provide citation for all methods used. |
||
Response 3: Thank you for mention, I’ve added missing citation to the methods section. |
||
Comments 4: L58.- What morbidity did the patients studied have? |
||
Response 4: Dear colleague, I apologize for possible misunderstanding, if I got your question correct, all patients that were included in this study were diagnosed with COVID-19, severity and lethality numbers in the studied group are presented in Tables 1 and 2. |
||
Comments 5: Limitations are missing. The paper does not include a section discussing the limitations of the study. |
||
Response 5: We’ve made additional Discussion paragraph about methods limitations. |
||
Comments 6: The authors must present a model that integrates the regulatory keys and signal transduction of the immune response, intercellular communication and metabolism. |
||
Response 6: Again, I have to apologize if there is any misunderstanding. We have used genexplain.com platform and its tool Genome Enhancer to perform a master regulator network analysis, it is mentioned in Methods section, and result networks can be found in Supplementary section. |
Reviewer 2 Report
Comments and Suggestions for Authors
- This research evaluates the transcriptome of samples from patients with COVID-19 under different conditions (lethality, severity, and the presence of cytokine storm).
- The topic is not entirely original, as previous studies are cited throughout the manuscript. However, the main contribution of this study is addressing a gap in the field by comparing different conditions using samples from the same hospital/patient group.
- As mentioned above, while the work may not introduce groundbreaking innovations, it provides a solid basis for comparing different conditions using samples from the same patients.
- To my knowledge, the methodology is adequate, and the authors explain all the necessary steps to achieve their results.
- Are the conclusions consistent with the evidence and arguments presented
and do they address the main question posed? Please also explain why this
is/is not the case. The conclusions are solely based on the findings, and they discuss the main results without exaggeration. However, the Conclusions section could be more succinct to avoid repeating the Discussion. - There are few references, but all of them cover the main topics and studies in the field.
- additional comments on the tables and figures. In Table 1, please clarify what F and M represent (e.g., by adding a note in the footnote). Cite each figure in the appropriate section of the Results rather than only at the end of the topic. Better introduce ALOX15, which was found to be differentially expressed in the results. The results presented as Appendix need be changed to Supplementary Material. The title could be more specific. The current title is generic and does not accurately convey the results presented in the manuscript.
Author Response
Response to Reviewer 2 Comments
|
||
1. Summary |
|
|
Thank you for dedicating your time to review this manuscript. Below, you will find our detailed responses, along with the relevant revisions and corrections included in the re-submitted files.
|
||
2. Point-by-point response to Comments and Suggestions for Authors |
||
Comments 1: In Table 1, please clarify what F and M represent (e.g., by adding a note in the footnote). |
||
Response 1: Thank you for mentioning, corresponding clarification was made at L76. |
||
Comments 2: Cite each figure in the appropriate section of the Results rather than only at the end of the topic. |
||
Response 2: Thanks, we’ve done the corrections in text. |
||
Comments 3: Better introduce ALOX15, which was found to be differentially expressed in the results. |
||
Response 3: I’ve expanded the part of the discussion section related to the ALOX15 gene. |
||
Comments 4: The results presented as Appendix need be changed to Supplementary Material. |
||
Response 4: Corresponding correction was made. |
||
Comments 5: The title could be more specific. The current title is generic and does not accurately convey the results presented in the manuscript. |
||
Response 5: We’ve revised and reconsidered the title of the article in accordance with your feedback. |
Reviewer 3 Report
Comments and Suggestions for Authors
1. The threshold for differentially expressed genes (p<0.01) is relatively loose, and it is recommended to use stricter correction methods (such as FDR) to reduce false positives.
2. When using non parametric tests (Mann Whitney), it is necessary to indicate whether the normality of the data has been tested. If the data follows a normal distribution, it is recommended to use t-test to improve sensitivity.
3. Paired sample analysis only identified 24 differentially expressed genes, which may result in insufficient sensitivity due to the small sample size (n=8), and its clinical significance should be interpreted with caution.
4. The prediction of main regulatory factors (such as CDH2 and FLT3) is based on network analysis, but lacks experimental validation (such as qPCR and protein level detection). Suggest supplementing in vitro experiments or citing published validation data.
During the discussion, it was mentioned that ALOX15 is associated with inflammation resolution, but key literature such as the Resolvin pathway study was not cited. Suggest supplementing relevant mechanism research to support the conclusion.
6. The age data in Table 1-4 (such as "60,9 ± 13,3") should be uniformly formatted with decimal points (60.9 ± 13.3)
Author Response
Response to Reviewer 3 Comments
|
||
1. Summary |
|
|
We appreciate you taking the time to review this manuscript. Below, you will find our comprehensive responses, along with the necessary revisions and corrections provided in the re-submitted files.
|
||
2. Point-by-point response to Comments and Suggestions for Authors |
||
Comments 1: The threshold for differentially expressed genes (p<0.01) is relatively loose, and it is recommended to use stricter correction methods (such as FDR) to reduce false positives. |
||
Response 1: Thank you for your comment regarding the threshold for differentially expressed genes. You are absolutely right, but even though the use of FDR reduces the number of false positive results, it also reduces sensitivity, especially if the data set is small. In studies with limited sample numbers or weak effects, the use of FDR may result in the loss of biologically relevant differentially expressed genes. We have relatively small sample size and no technical repeats, so we decided to lower the p-value threshold to 0.01 instead of applying FDR correction. |
||
Comments 2: When using non parametric tests (Mann Whitney), it is necessary to indicate whether the normality of the data has been tested. If the data follows a normal distribution, it is recommended to use t-test to improve sensitivity. |
||
Response 2: Thanks, we’ve tested our distributions with Shapiro-Wilk normality test and results were negative, that is why we’ve decided to use nonparametric test. I’ve added corresponding in Statistical Analysis subsection of Methods section. |
||
Comments 3: Paired sample analysis only identified 24 differentially expressed genes, which may result in insufficient sensitivity due to the small sample size (n=8), and its clinical significance should be interpreted with caution. |
||
Response 3: Thanks, I’ve done the corrections in text. |
||
Comments 4: The prediction of main regulatory factors (such as CDH2 and FLT3) is based on network analysis, but lacks experimental validation (such as qPCR and protein level detection). Suggest supplementing in vitro experiments or citing published validation data. |
||
Response 4: Thank you for this comment, we’ve included additional paragraph in discussion with limitations of methods we used. |
||
Comments 5: During the discussion, it was mentioned that ALOX15 is associated with inflammation resolution, but key literature such as the Resolvin pathway study was not cited. Suggest supplementing relevant mechanism research to support the conclusion. |
||
Response 5: Thanks, I’ve added additional information and citation bout ALOX15 in the Discussion section. |
||
Comments 6: The age data in Table 1-4 (such as "60,9 ± 13,3") should be uniformly formatted with decimal points (60.9 ± 13.3) |
||
Response 6: Thank you for mention, I’ve made corresponding correction. |
Round 2
Reviewer 1 Report
Comments and Suggestions for Authors
Response 4: While my previous question may have been unclear, I believe incorporating an analysis of comorbidity frequency from the hospital's biobank collection would substantially enrich your research. The established link between comorbidities and COVID-19 severity makes this a crucial aspect to consider. Tables 1 and 2 would benefit greatly from the inclusion of this data. DOI: 10.1021/acsptsci.2c00181, DOI: 10.14336/AD.2020.0619, DOI: 10.3389/fimmu.2023.1111797
Response 6: The interoperable tools and databases within Genome Enhancer (TRANSPATH®, CellChat, Reactome, HumanCyc) provide a strong foundation for building integrative models of immune regulation, communication, and metabolism. Utilizing its capacity to link enhancer biology with systems-level pathway analysis can yield significant findings. To effectively communicate these findings, the authors should create a figure that summarizes the most important results. This will substantially improve the overall quality of their work, provided the data is reliable and the workflow is appropriately customized.
Comments on the Quality of English LanguageNone
Author Response
Response to Reviewer 1 Comments
|
||
1. Summary |
|
|
Thank you for dedicating your time to review this manuscript. Below, you will find our detailed responses, along with the relevant revisions and corrections included in the re-submitted files.
|
||
2. Point-by-point response to Comments and Suggestions for Authors |
||
Comments 4: While my previous question may have been unclear, I believe incorporating an analysis of comorbidity frequency from the hospital's biobank collection would substantially enrich your research. The established link between comorbidities and COVID-19 severity makes this a crucial aspect to consider. Tables 1 and 2 would benefit greatly from the inclusion of this data. DOI: 10.1021/acsptsci.2c00181, DOI: 10.14336/AD.2020.0619, DOI: 10.3389/fimmu.2023.1111797 |
||
Response 4: Thank you for your detailed reply! We included information about comorbidity in Tables 1 and 2 in a form of Charlson comorbidity index calculated for the following diseases: history of myocardial infarction, congestive heart failure, peripheral arterial disease, cerebrovascular disease, dementia, chronic lung disease, connective tissue disease, peptic ulcer disease, diabetes mellitus, kidney damage, hemiplegia or paraplegia, leukemia, lymphoma, malignancy, liver damage, and AIDS. Concerning comorbidity analysis: recently we have published the results of the metabolomics investigation with the same material from the same patients (DOI: 10.1038/s41598-025-90426-0). One of results was that the differences between the comorbid and non-comorbid groups of COVID-19 patients with cytokine storm were not significant at the metabolomic level. We understand that it is greatly indirect evidence of the functioning of transcriptome in the severe cases with comorbidity but now we have not another data of influencing comorbidity on the severity in our samples, and it was not the main goal of this study. |
||
Comments 6: The interoperable tools and databases within Genome Enhancer (TRANSPATH®, CellChat, Reactome, HumanCyc) provide a strong foundation for building integrative models of immune regulation, communication, and metabolism. Utilizing its capacity to link enhancer biology with systems-level pathway analysis can yield significant findings. To effectively communicate these findings, the authors should create a figure that summarizes the most important results. This will substantially improve the overall quality of their work, provided the data is reliable and the workflow is appropriately customized. |
||
Response 6: Dear reviewer! We apologized and agree with you that such model will improve the work in whole. Additional summary picture was added at the end of the Results section |
Reviewer 3 Report
Comments and Suggestions for Authors
The authors have answered my questions well. Thank you for your efforts.
Author Response
Comments 1: The authors have answered my questions well. Thank you for your efforts. |
Response 1: Dear reviewer, thank you very much for revising our work! |